# Performance Characterization of a Fully Transportable Mid-Infrared Laser Heterodyne Radiometer (LHR)

**DOI:** 10.3390/s23020978

**Published:** 2023-01-14

**Authors:** Fengjiao Shen, Xueyou Hu, Jun Lu, Zhengyue Xue, Jun Li, Tu Tan, Zhensong Cao, Xiaoming Gao, Weidong Chen

**Affiliations:** 1School of Advanced Manufacturing Engineering, Hefei University, Hefei 230601, China; 2Laboratoire de Physico-Chimie de l’Atmosphère, Université du Littoral Côte d’Opale, 59140 Dunkerque, France; 3Anhui Institute of Optics and Fine Mechanics, Chinese Academy of Sciences, Hefei 230031, China

**Keywords:** laser heterodyne radiometer, polycrystalline mid-infrared fiber, fully transportable

## Abstract

A fully transportable laser heterodyne radiometer (LHR), involving a flexible polycrystalline mid-infrared (PIR) fiber-coupling system and operating around 8 µm, was characterized and optimized with the help of a calibrated high temperature blackbody source to simulate solar radiation. Compared to a mid-IR free-space sunlight coupling system, usually used in a current LHR, such a fiber-coupling system configuration makes the mid-infrared (MIR) LHR fully transportable. The noise sources, heterodyne signal, and SNR of the MIR LHR were analyzed, and the optimum operating local oscillator (LO) photocurrent was experimentally obtained. The spectroscopic performance of the MIR LHR was finally evaluated. This work demonstrated that the developed fully transportable MIR LHR could be used for ground-based atmospheric sounding measurements of multiple trace gases in the atmospheric column. In addition, it also has high potential for applications on spacecraft or on an airborne platform.

## 1. Introduction

Measurement of the vertical concentration profiles of atmospheric trace gases is very important for understanding the physics, chemistry, dynamics, and radiation budget of the atmosphere as well as validating chemical models and satellite observations [1,2,3,4]. Laser heterodyne radiometer (LHR), as a kind of passive remote sensing technology, was introduced and developed in the 1970s to meet the needs of observing the ozone hole in the atmosphere [5,6,7]. Since then, LHR applications have been virtually silent, as a result of the lack of a suitable tunable laser source as the local oscillator (LO) for heterodyne measurements [8]. In the past 10 years, LHR technology has been revived due to significant advances in laser and photonic technology. Compared to the currently used ground-based Fourier-transform spectrometer (FTS) for measuring atmospheric trace gases, LHR is advantageous in terms of high spectral resolution (<0.001 cm^−1^), high sensitivity (within a factor of ~2 of the quantum noise limit), and high spatial resolution due to its very small coherent field of view and cost-effective compact instrument size [8,9].

Near-infrared (NIR) LHR has been growing in recent years. Wilson et al. [10,11,12], Rodin et al. [13,14], and Wang et al. [15] developed LHR using a telecom-grade distributed feedback (DFB) laser as LO for remote sensing CO_2_ and CH_4_ near 1.6 μm and 1.65 µm in the atmospheric column. In the NIR range, progressive fiber photonic devices enable compact, portable, and optically aligned LHRs [8]. The atmospheric windows of 3–5 μm and 8–12 μm correspond to the strong fundamental ro-vibrational absorptions of a large number of key atmospheric species, which are more attractive for LHR-sensitive remote sensing [9]. A dual-channel LHR involving two interband cascade lasers (ICL) centered at 3.53 µm and 3.93 µm as LOs has been recently reported for remote sensing atmospheric H_2_O vapor, N_2_O, and CH_4_ [16,17,18]. Weidmann et al. developed an LHR based on an external cavity quantum cascade laser (EC-QCL) [19] as the LO for working in the 8.08–8.93 µm range [20,21], in which O_3_, N_2_O, CH_4_, CCl_2_F_2_, and H_2_O have been measured. In their reported works, a non-removable sun tracker was installed on the roof terrace to collect sunlight, limiting their LHR’s potential for field applications [8].

In this paper, we report the performance characterization of a fully transportable MIR LHR using a calibrated blackbody (BB) source. A newly commercially available polycrystalline mid-IR (PIR) fiber is exploited to couple the solar or BB radiation into the LHR receiver, which renders the MIR LHR fully transportable. In the subsequent sections, details of the study are presented in the following order: (1) instrumental description of the fully transportable MIR LHR; (2) performance characterization of the MIR LHR, including (i) the signal to noise ratio (SNR) of the LHR system containing (a) noise sources, (b) theoretical SNR vs. measured SNR, and (c) the heterodyne signal and SNR of the LHR vs. the LO photocurrent and (ii) the spectroscopic performance of the LHR system containing (a) the impact of the RF pass-band on the LHR spectrum and (b) the LHR spectral line shape vs. the scanning speed of the LO frequency; (3) conclusion and future work.

## 2. Instrumental Description

The schematic of a fully transportable MIR LHR involving a PIR fiber-coupling solar or BB radiation system is shown in Figure 1. A portable EKO sun-tracker is used for the development of a fully transportable MIR LHR. A 1.5 m long PIR fiber made of AgClBr (PIR900/1000-150-TI/SMA-TI/SMA-PE32, Art Photonics Inc. Berlin, Germany) is employed to couple the solar or BB radiation to the LHR receiver with a numerical aperture (NA) of 0.3 (with a fiber core diameter of 860 µm). This is transparent across a broad spectral range of 3–18 μm, with a transmission of ~54% at 8 µm. This fiber, with a minimum bend radius of 150 mm, is very flexible and suitable for coupling cw laser power of up to 40 W and is capable of operating over a temperature range of −50 to 140 °C without an aging effect [8]. The PIR fiber-coupling system, installed on the sun-tracker, includes lens 1 (in CaF_2_), which is associated with a Ge filter to filter radiation out of the spectral region of interest to minimize the shot noise resulting from the signal source [8]. Sunlight is focused by lens 1 into a PIR fiber. The beam emerging from the fiber is collimated with lens 2 (in Germanium); the amplitude is modulated at 1 kHz using a mechanical chopper (Model MC2000B, Thorlabs, Inc. Newton, NJ, USA) and then directed to a 5% (R)–95% (T) beam splitter (BS, in CaF_2_). The solar beam is superimposed with the LO beam from the EC-QCL (TLS-41000-MHF, Daylight solutions Inc. San Diego, CA, USA). The EC-QCL is controlled with a specific controller (Model 1001-TLC, Daylight solutions Inc.) and is continuously tunable at a frequency from 1223 to 1263 cm^−1^ (7.91–8.17 µm) through its external-cavity tuning. The laser provides optical output powers of more than 60 mW over 40 cm^−1^ spectral coverage with a power fluctuation less than 1% r.m.s (root mean square) over 5 min, according to the manufacturer. The LO light is size-expanded by a factor of three through two OAPMs (OAPM1 and OAPM2). It is intensity-controlled and polarization-adjusted using a wideband IR polarizer (2–12 µm). The combined solar and LO beams are then focused by an OAPM3 onto a VIGO photomixer (PVI-4TE-10.6, VIGO System S.A. Ożarów Mazowiecki, Poland). The radio frequency (RF) beat note signal from the photomixer enters the RF receiver through a 27.5–270 MHz band-pass filter, followed by two-stage low noise amplifiers (Model ZFL-500LN+, Mini-Circuits^®^ New York, NY, USA) with a gain of 24 dB each. After that, the amplified beat signal is connected to a square-law Schottky diode (Model 8472B, Keysight Inc. Colorado Springs, CO, USA) for a power measurement of the RF signal, which is proportional to the square of the input beat signal amplitude. The output signal from the square-law detector is demodulated with a lock-in amplifier (LIA, Model 5110, Ametek Inc. Berwyn, PA, USA). A National Instruments data-acquisition card (PCI 6251, NI Inc. Austin, TX, USA) is used to digitalize the output signal from the LIA via a LabVIEW program. The data are then transferred to a laptop for further data processing.

One part of the LO beam (transmitted LO power from the BS) is directed to either a 12.5 cm single-pass cell filled with a CH_4_ mixture in air for absolute frequency determination or to a 12.5 cm long home-made Ge etalon (free spectral range (FSR) = 0.0333 cm^−1^) for relative frequency calibration when the LO frequency is tuned.

In order to evaluate and optimize the MIR LHR’s performance, a calibrated BB source (Model 67030, Newport Inc. Florida, USA) held at 1000 °C is used. Here, the source temperature changes the source power and, thus, the power of the heterodyne signal. However, the temperature difference has no effect on the characteristics of the heterodyne signal. The BB radiation is collected and collimated with lens 3 (in CaF_2_) and then coupled to the PIR fiber through a second lens, lens 4 (in CaF_2_). The BB radiation emerging from the fiber is then collimated and injected into the LHR receiver via lens 2.

## 3. Performance Characterization of the MIR LHR

### 3.1. SNR of the MIR LHR

In order to characterize the MIR LHR’s performance using a PIR fiber-coupling system for sunlight collection, investigation in terms of the SNR of the LHR signal was first performed using the calibrated BB source. In this study, time-series measurements of the RF signal from the LIA with and without LO radiation injection were carried out. The LO and BB power-induced photocurrents, were 610 µA and 117 µA, respectively. A time constant of 1 s combined with low-pass filters (12 dB/oct) was used for the LIA.

Based on the measurement of a beat note amplitude of 150.18 µV and a noise level (1 σ) of 0.36 µV in the beat note (Figure 2), a SNR of ~419 was deduced.

The theoretical shot-noise-limited SNR of the LHR can be expressed by Equation (1) [22,23,24]. This equation can be used to assess the experimental LHR performance when referenced to the ideal shot-noise limit. In other words, it corresponds to the expected performance of an ideal LHR in which shot noise is the sole noise [25]:(1)$$\mathrm{SNR}=\eta_{het}\cdot \frac{\eta \cdot P_{S}\cdot \sqrt{B\cdot \tau }}{h\cdot \nu \cdot B}=\eta_{het}\cdot \frac{\eta \cdot \kappa \cdot \sqrt{B\cdot \tau }}{exp\left(\frac{h\cdot \nu }{k\cdot T}\right)-1}$$
where the overall optical coupling efficiency *κ* is defined as the fraction of the incoming signal radiation transmitted through all the optical components to the photomixer [25]. For the BB, this accounts for two CaF_2_ lenses (transmission ~0.92 each), the PIR fiber (transmission ~0.54), the Germanium lens (transmission ~0.99), the BS (transmission ~0.95), the silver-plated OAPM (assumed reflectance ~0.97), and the BaF_2_ wedged window in front of the photomixer (transmission ~0.95), resulting in an overall coupling efficiency of 0.396. The heterodyne efficiency *η_het_* of the photomixing of the sunlight beam and the LO beam is considered to be 1 in an ideal case [26]; the quantum efficiency *η* of the photomixer provided by the manufacturer at this wavelength is 0.5. Under these conditions, given an LHR double-side bandwidth of 485 MHz (for an RF pass-band of 27.5–270 MHz) and a 1 s time constant of the LIA at an LO frequency (1242.3 cm^−1^ or 3.73 × 1013 Hz), a shot-noise-limited SNR of about 1430 was obtained. The theoretical shot-noise-limited SNR is 3.4 times higher than the measured SNR at a photocurrent of 610 µA, or a factor *ρ* of 0.29 in the theoretical SNR was observed.

#### 3.1.1. Noise Sources in the MIR LHR

In order to understand the difference between the measured and the calculated ideal SNRs, the noises in various parts of the MIR LHR were investigated with the help of a signal analyzer (N9000B, CXA X-Series, Keysight, Inc. Newton, MA, USA). In this study, the total noise density *N_total_* ($\mathrm{nV}/\sqrt{Hz}$) of the MIR LHR can be expressed by [8,9]
(2)$$N_{total}=\sqrt{N_{Det}^{2}+N_{LIN}^{2}+N_{SIN}^{2}}$$
(3)$$N_{LIN}^{2}=N_{LSN}^{2}+N_{LEN}^{2}$$
(4)$$\mathrm{and}\;N_{SIN}^{2}=N_{SSN}^{2}+N_{SEN}^{2}$$
where *N_Det_*, *N_LIN_*, and *N_SIN_* are the detection chain noise, the laser-induced noise, and the signal-induced noise, respectively. The laser-induced noise *N_LIN_* includes laser-induced shot noise *N_LSN_* and laser excess noise *N_LEN_*, respectively, while the signal-induced noise *N_SIN_* includes signal-induced shot noise *N_SSN_* and signal excess noise *N_SEN_*.

The contribution of each type of noise in the above expressions is discussed below in detail.

(1)Detection chain noise (*N_Det_*)

The detection chain noise *N_Det_* includes the background noise, the Johnson noise resulting from the amplifier associated with the photodetector, and the dark noise [8]. It can be measured when there is no light incident on the detector.

In order to identify noise sources, the signals at the RF output of the photomixer were measured with the signal analyzer under different conditions: (A) with the input of the signal analyzer directly connected to a resistance of 50 Ω (Figure 3, blue curve) for the measurement of the background noise *N_SA_* of the signal analyzer; (B) with only the detector output connected to the signal analyzer (Figure 3, red curve), representing the sum of the signal analyzer’s background noise N_SA_ and the detection chain noise *N_Det_*; (C) with the presence of the LO but without the BB radiation injection (Figure 3, green curve), representing the sum of the signal analyzer’s background noise *N_SA_*, the detection chain noise *N_Det_*, and the laser-induced noise *N_LIN_*; (D) with the presence of both LO and BB radiations (Figure 3, black curve), representing the sum of the heterodyne signal, the signal analyzer’s background noise *N_SA_*, the detection chain noise *N_Det_*, the laser-induced noise *N_LIN_*, and the BB source-induced noise *N_SIN_*.

Comparing the RF output (after the associated amplifier to the photodetector) of the photomixer with both the LO and BB radiation injections (Figure 3, black curve) to that of the photomixer with only the LO injection (Figure 3, green curve), the two signals were almost overlapped, which indicates that there was almost no additional noise contribution from the BB radiation, i.e., signal-induced noise *N_SIN_* can be neglected at this stage of analysis.

Therefore, the total noise density *N_total_* (Equation (2)) of the LHR system becomes
(5)$$N_{total}=\sqrt{N_{Det}^{2}+N_{LIN}^{2}}$$

As can be seen in Figure 3, the LHR system noise (measured at the RF output of the photomixer) was mainly distributed in the range of 1 kHz–100 MHz. The noise is mainly composed of 1/*f* and white noises. In the low-frequency range, the noise is dominated by 1/*f*, and then the noise distribution tends to be flat. Therefore, the pass-band should be selected in the higher-frequency range where 1/*f* is significantly reduced. In order to study the characteristics of white noise in the high-frequency range, the detection chain noise *N_Det_* was analyzed at 150 MHz, where the noise is dominated by white noise.

Given the readings from the signal analyzer at 150 MHz, 0.00709 mV for the noise amplitude of the signal analyzer’s background noise (Figure 3, blue curve) and 0.02370 mV for the noise amplitude (Figure 3, red curve), resulting from the signal analyzer background noise and the detection chain noise, respectively, the detection chain noise amplitude of 0.02261 mV can be deduced. This was the integrated detection chain noise level within the set resolution bandwidth (RBW) of 100 kHz; the corresponding detection chain noise density *N_Det_* was, thus, 71.5 $\mathrm{nV}/\sqrt{Hz}$.

Comparing the signal levels with the LO injection (Figure 3, green curve) and without the LO injection (Figure 3, black curve), an increase in the noise level was observed, which primarily indicates an influence by the LO injection, including laser-induced shot noise and laser excess noise, both resulting from the EC-QCL LO that was used.

Next, laser-induced noise *N_LIN_*, including laser-induced shot noise *N_LSN_* and laser excess noise *N_LEN_*, was analyzed.

(2)Laser-induced noise (*N_LIN_*)

In order to quantitatively analyze the contributions of laser-induced shot noise N_LSN_ and laser excess noise N_LEN_, a detailed study was carried out. The LO power injected into the photomixer was adjusted through a polarizer in front of the LO laser, and the LO power-induced photocurrent was used as the indicator of the LO power injected into the photomixer.

Noise from the RF output of the photomixer within its pass-band of 1 kHz–500 MHz was analyzed using the signal analyzer, as shown in Figure 4 upper, at different LO photocurrents (note: here the signal analyzer background noise has all been subtracted).

The system total noise density *N_total_* in [$nV/\sqrt{Hz}$] (Figure 5, black square) was calculated by dividing the mean of all the total noise amplitudes (including detection chain noise *N_Det_* and laser-induced noise *N_LIN_*) measured in the pass-band of 27.5–270 MHz (i.e., the pass-band of the used RF filter) (Figure 4 lower) by the square root of the RBW (100 kHz). Given the detection chain noise density *N_Det_* of 71.5 $\mathrm{nV}/\sqrt{Hz}$ (Figure 5, gray line) determined above, the laser-induced noise (Figure 5, blue triangle) was calculated by subtracting the detection chain noise *N_Det_* (Figure 5, gray line) from the system total noise density *N_total_* (Figure 5, black square), based on Equation (5).

Based on the laser-induced noise *N_LIN_* deduced as shown in Figure 5, the laser-induced shot noise *N_LSN_* and laser excess noise *N_LEN_* are studied in detail.

(2.1)Laser-induced shot noise (*N_LSN_*)

The voltage *V_SN_* and current *I_SN_* of the laser-induced shot noise can be calculated using Equations (6) and (7), respectively [27]:(6)$$V_{SN}\left(V\right)=R_{ti}\cdot I_{SN}\left(A\right)$$
(7)$$I_{SN}\left(A\right)=\sqrt{2\cdot e\cdot I_{DC}\left(\mu A\right)\cdot 10^{-6}\cdot B}$$
(8)$$\mathrm{and}\;I_{DC}\left(\mu A\right)=V_{DC}\left(V\right)\cdot 10^{6}/R_{ti}$$
where *R_ti_* = 6000 *V/A* is the transimpedance of the photomixer, *e =* 1.602 × 10^−19^ C is the electric charge, *B* = 5 × 10^8^ Hz is the bandwidth of the photomixer over which the noise is considered, *I_DC_* is the DC LO photocurrent induced by the LO laser power in the photomixer, *V_DC_* is the DC LO voltage induced by LO laser power monitored at the output of the photomixer, and 10^6^ represents the unit conversion from *V* to *µV*.

From Equations (6)–(8), the laser-induced shot-noise density *N_LSN_* (in [$nV/\sqrt{Hz}$]) can be expressed as [28]
(9)$$N_{LSN}\left(nV/\sqrt{Hz}\right)=3.396\cdot \sqrt{I_{DC}\left(\mu A\right)}$$

The calculated laser-induced shot-noise density *N_LSN_* is plotted in Figure 5 (red circle). 

(2.2)Laser excess noise (*N_LEN_*)

The laser excess noise *N_LEN_* (Figure 5, green star) can then be calculated using Equations (3) and (9), that is, by subtracting the laser-induced shot noise *N_LSN_* (Figure 5, red circle) from the laser-induced noise *N_LIN_* (Figure 5, blue triangle).

As can be seen from Figure 5, at LO photocurrents I_DC_ < 450 µA, the detection chain noise *N_Det_* (gray line) is larger than the laser-induced shot noise *N_LSN_* (red circle); however, only in the range of 450 µA < I_DC_ < 656 µA are the laser excess noise *N_LEN_* (green star) and the detection chain noise *N_Det_* (gray line) both smaller than the laser-induced shot noise *N_LSN_* (red circle), so the LHR system operates in a shot-noise dominated regime (with a shot-noise-limited high-sensitivity performance). The higher LO photocurrent limit could not be verified, because saturation of the detector preamplifier for optical powers above 733 µA was observed.

#### 3.1.2. Theoretical SNR vs. Measured SNR

Based on the above analysis of the LHR system noises, when comparing the measured SNR (~419) to the theoretical SNR (~1430), the factor of 0.29 (=419/1430) is analyzed as follows:

(1) At an LO power-induced photocurrent of 610 µA, at which SNR = 419 was determined, the laser excess noise *N_LEN_* (Figure 5, green star) and the detection chain noise *N_Det_* (Figure 5, gray line) were both 0.9 times the laser-induced shot noise *N_LSN_* (Figure 5, red circle). The total noise of the LHR system *N_total_*, thus, exceeds the laser-induced shot noise *N_LSN_*, using Equations (2)–(4), by
(10)$$N_{total}=\sqrt{N_{Det}^{2}+N_{LSN}^{2}+N_{LEN}^{2}}=1.62\cdot N_{LSN}$$

The system total noise *N_total_* is 1.62 times the laser-induced shot noise *N_LSN_*, _and_ a correction factor *ρ*_1_ of 0.62 (=1/1.62) is obtained for the theoretically calculated SNR under the ideal shot-noise condition.

(2) The overall optical coupling efficiency *κ* of 0.40 for the BB radiation, based on the transmission of each optical component given by the manufacturer, would be overestimated. The measured overall optical coupling efficiency was about 0.27, and this difference in κ, between 0.27 and 0.40, may be due to the long-time exposure of these optical components to the humid and salty environment of the seaside. An additional correction factor *ρ*_2_ of 0.68 (=0.27/0.40) was obtained.

(3) The heterodyne efficiency *η_het_* of 1 for the ideal case would be overestimated. The difference in the shaped beam sizes of sunlight (12.5 mm) and the LO light (7.5 mm) caused a heterodyne efficiency of ~0.8 [9]. In addition, the photomixer has a small photo-sensitive area of 0.5 × 0.5 mm^2^, so even a small mismatched angle would result in a heterodyne efficiency of ~0.9 [9]. An additional correction factor *ρ*_3_ of 0.72 (=0.8 × 0.9) was achieved.

Based on the above analysis, the total correction factor *ρ*_exp_ for the theoretical SNR, resulting from the above reasons, is
*ρ*_exp_ = *ρ*_1_·*ρ*_2_·*ρ*_3_ = 0.30(11)

This value is comparable to the factor *ρ* of 0.29.

#### 3.1.3. Heterodyne Signal and SNR of the LHR vs. LO Photocurrent

Based on the study performed on the noises as the function of the LO power-induced photocurrent, the heterodyne signal and the corresponding SNR, as the function of the LO photocurrent, were investigated. The LO power was varied by adjusting a polarizer in front of the LO output, which both the mean heterodyne signal and its standard deviation were measured from, and the corresponding SNR was derived. The results are shown in Figure 6.

Figure 6 clearly shows that the amplitude of the heterodyne signal increases almost linearly at a low LO photocurrent (black square), which is consistent with $P_{HET}\propto P_{S}\times P_{L}$, before reaching a rollover point at an LO photocurrent of ~610 µA. The SNR increases with the LO power and reaches a maximum when the LO photocurrent is also equal to 610 µA. Above 610 µA, the heterodyne signal becomes smaller, and the noise on the heterodyne signal rapidly increased (c.f. Figure 5), that is, the SNR significantly decreased. The data show that the maximum SNR of the LHR can be obtained with an optimum LO photocurrent. The optimum LO photocurrent is highly dependent on the technology and characteristics of the detector (which is used as the photomixer) [25]. For each type of detector, the optimum LO photocurrent must be determined because saturation effects, detection chain noise, and detector response homogeneity will affect the performance.

Therefore, the LO power should be precisely adjusted in order to significantly reduce the noise on the heterodyne signal to achieve a high SNR, without any device saturation.

### 3.2. Spectroscopic Performance of the LHR System

Under the optimal LO photocurrent of ~610 µA, the spectroscopic performance of the LHR prototype was evaluated. To that end, the BB radiation was coupled to a direct absorption cell through the PIR fiber.

#### 3.2.1. Impact of RF Pass-Band on LHR Spectrum

To investigate the impact of the RF filter pass-band on the LHR spectral signal, the heterodyne absorption spectra of CH_4_ in a single-pass cell at 280 mbar were experimentally extracted and analysed with different pass-bands of the RF filters. In this study, the LO laser frequency was scanned at around 1242.00 cm^−1^, with a frequency scanning rate of 500 mHz; the LIA sensitivity and time constant were set to 5 mV and 10 ms, respectively.

LHR noise was investigated at first. Figure 7 shows the noise amplitude spectrum of an LHR signal from the RF output of the photomixer recorded using a signal analyzer (with a RBW of 100 kHz). As can be seen, there are strong 1/*f* noises in the low-frequency range, which have to be filtered out.

The noise densities within the different pass-bands of the RF filters available in the present work were estimated by dividing the integral noise over each pass-band by both the bandwidth and the square root of the RBW (100 kHz) (Table 1). As shown in Table 1, the noise density of the RF filter with a pass-band of 27.5–270 MHz was the lowest, and the corresponding SNR was the highest because of its wider filter bandwidth.

Using a PZT, driven by a sine-wave signal (with an amplitude of +3.0 V_PP_ and an offset of +1.6 V_DC_), to scan the external cavity (EC) of the LO laser and, hence, the laser frequency, the LHR CH_4_ absorption spectra were then recorded with the five available RF filters, as shown in Figure 8. The LHR CH_4_ absorption spectra were measured with AC-coupling band-pass RF filters (Figure 8 left), which can significantly filter out low frequency noises, and had a higher SNR and better spectral line shape, while the spectra measured with the DC-coupling low-pass RF filters (Figure 8 right) are noisy, resulting from high 1/*f* noise.

The noise level of the spectrum measured with the RF filter with a pass-band of 27.5–270 MHz (Figure 8 left, black) is lower than that measured with a pass-band of 27–33 MHz (Figure 8 left, red). Moreover, there is no obvious difference in the spectral line widths between the two LHR CH_4_ absorption spectra. The reason is that the theoretical spectral line width (FWHM) of the CH_4_ absorption around 1242.00 cm^−1^ at 280 mbar is 0.034 cm^−1^ (1020 MHz [29,30,31,32], which is wider than the doubled RF filter bandwidths (12 MHz and 485 MHz); therefore, there is not any obvious effect of the RF filter bandwidths on the measured line widths.

Based on such an analysis, the RF filter with a pass-band of 27.5–270 MHz was selected for the current LHR system.

#### 3.2.2. LHR Spectral Line Shape vs. Scanning Speed of LO Frequency

In order to obtain a high-precision heterodyne spectrum (with a minimized line shape shift and distortion), the scanning time Δ*T_scan_* across the halfwidth of the absorption line Δ*ν_HWHM_* (in (cm^−1^)), which is also dependent on the scanning speed of the LO frequency *υ_sc_* (in (cm^−1^/s)), should be appropriately adapted to an LIA time constant τ, such that [33]
(12)$$\Delta T_{scan}=\Delta \nu_{HWHM}/\upsilon_{SC}\geq 14\cdot \tau$$

An experimental investigation on the BB-based heterodyne spectra using different scanning speeds with a selected LIA time constant of 1 ms was carried out. In this study, the LO laser frequency was scanned at around 1242.00 cm^−1^; the LIA sensitivity was set to 2 mV; and the pressure of the CH_4_ mixture in the air in the single-pass cell was about 280 mbar.

The LHR CH_4_ absorption spectra recorded by scanning the LO frequency at different speeds (see Table 2), using the same LIA time constant of 1 ms, are plotted in Figure 9. As can be seen, the selection of the scanning speed of the LO frequency should be well-matched to the applied LIA time constant. Comparing these four heterodyne absorption spectra, the linewidth broadening Δ*ν_b_* (FWHM), line shift δ*ν*, and noise level (1σ) of these spectra were analyzed based on their Voigt profile fit (Figure 9, blue) and summarized in Table 2. Given the combination of the 1 ms time constant and the 0.3935 cm^−1^/s scanning speed (with a scanning time of 36 ms), the heterodyne spectrum has a lower noise level and a smaller line shift, while keeping a high spectral resolution (Figure 9b and Table 2).

Meanwhile, it is generally considered that the scanning speed of the LO frequency υ_sc_ being set to a slower value, which meets Equation (12) well, can limit the distortion of the line shape.

As can be seen from Table 2, at an LO frequency scanning speed of 0.1574 cm^−1^/s, the scanning time of 89 ms was far more than 14 times the applied LIA time constant of 1 ms. In this case, the measured line shift was the smallest, the absorption depth was the deepest, and the linewidth Δ*ν_b_* (Table 2) was the narrowest and close to the theoretical value of ~0.036 cm^−1^ [29,30,31,32]. However, the noise (1 *σ*: 0.0091) was the highest (Table 2). Hence, a trade-off has to be made. Therefore, at the LIA time constant of 1 ms, when the scanning speed of the LO frequency was 0.3935 cm^−1^/s (the scanning time of 36 ms), for which the noise (1 σ: 0.0043) was relatively low, the line shift (~0.0004 cm^−1^) was relatively small, and the achievable spectral resolution (spectral line width of ~0.042 cm^−1^) was relatively high. Thus, this was selected for the LHR measurement.

## 4. Conclusions

In this paper, the evaluation and optimization of the fully transportable MIR LHR, with the help of a stable high temperature BB source to simulate the sunlight, was performed. The noise sources, heterodyne signal, and SNR of the MIR LHR were analyzed, and the optimum operating LO photocurrent was experimentally obtained. The spectroscopic performance of the MIR HR was finally evaluated. This work demonstrated that the developed transportable MIR LHR could be used for ground-based atmospheric sounding measurements of trace gases in the atmospheric column. In future work, the developed LHR prototype will be tested and validated via ground-based field measurements of the tropospheric trace gases in the atmospheric column, and vertical profiles of the trace gases in the atmospheric column will be retrieved from the ground-based measured LHR spectrum using a retrieval algorithm, given a decent knowledge of the additional atmospheric parameters such as the solar angle, atmospheric temperature, atmospheric pressure, composition of the atmosphere, and refractive index [10]. In addition, this developed LHR prototype also has a high potential for applications on spacecraft or on an airborne platform. For the ongoing developments, we propose to develop a next generation LHR (NexLHR) based on the current, modern photonic integrated circuits (ICs) technology. The objective is to realize an “on-chip” LHR receiver. This will make the NexLHR cost-effective, energy efficient, autonomous, and ultra-portable, which will be most suitable for field deployment to measure the atmospheric vertical profiles of key atmospheric species through in situ ground-based measurements, aircraft, and satellite observations as well as to be integrated into the currently operational worldwide observation networks. Meanwhile, the proposed infrared LHR instrument presents a high potential for technology transfer and commercialization.

## Figures and Tables

**Figure 1 sensors-23-00978-f001:**
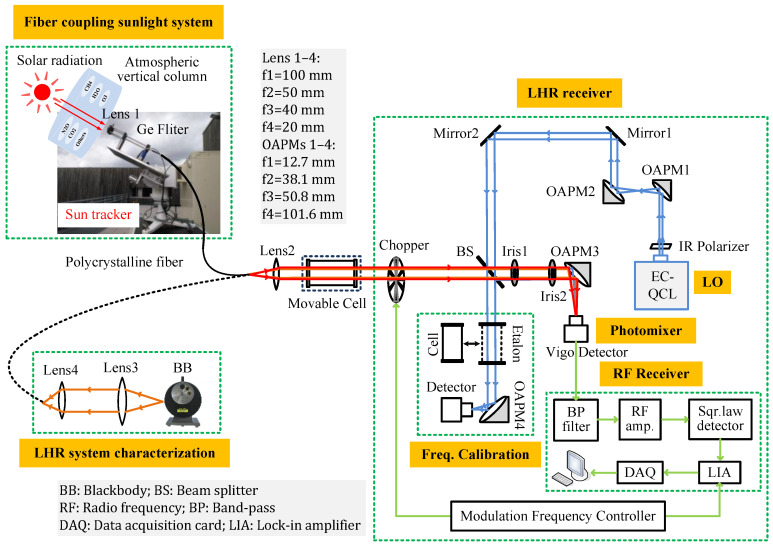
Schematic of a fully transportable LHR involving a PIR fiber-coupling solar or BB radiation system.

**Figure 2 sensors-23-00978-f002:**
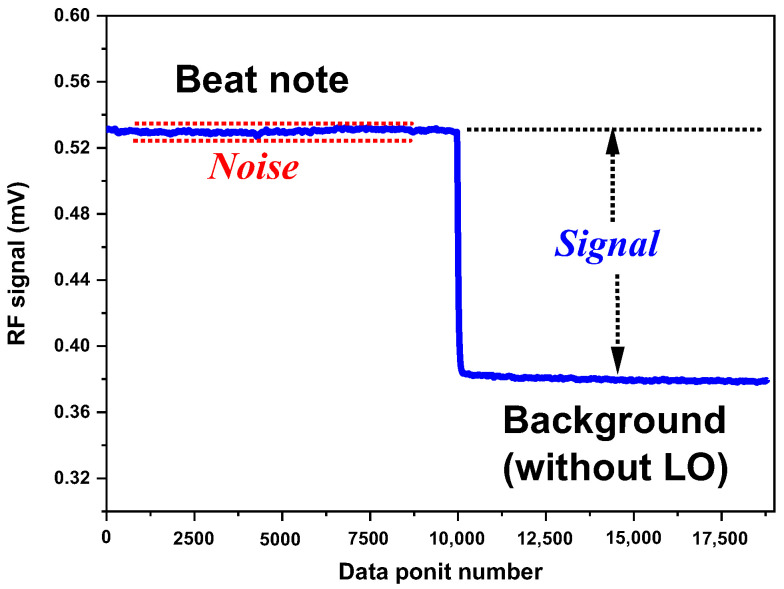
SNR measurement of the beat note at the output of the LIA.

**Figure 3 sensors-23-00978-f003:**
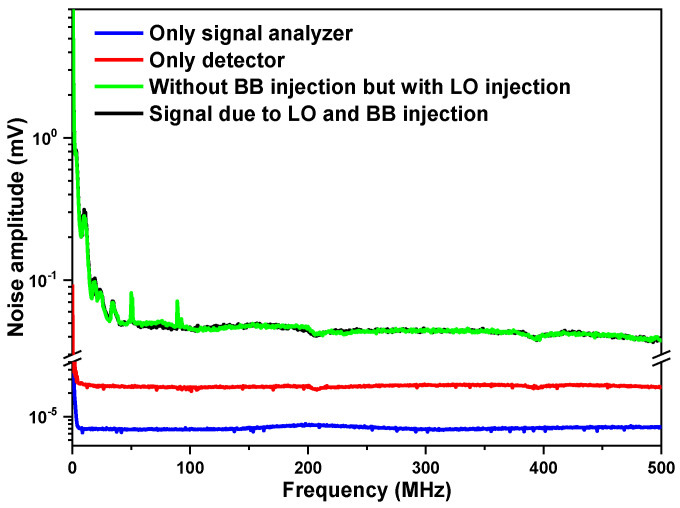
Noise amplitude spectra of the heterodyne system in the bandwidth of 1 kHz–500 MHz.

**Figure 4 sensors-23-00978-f004:**
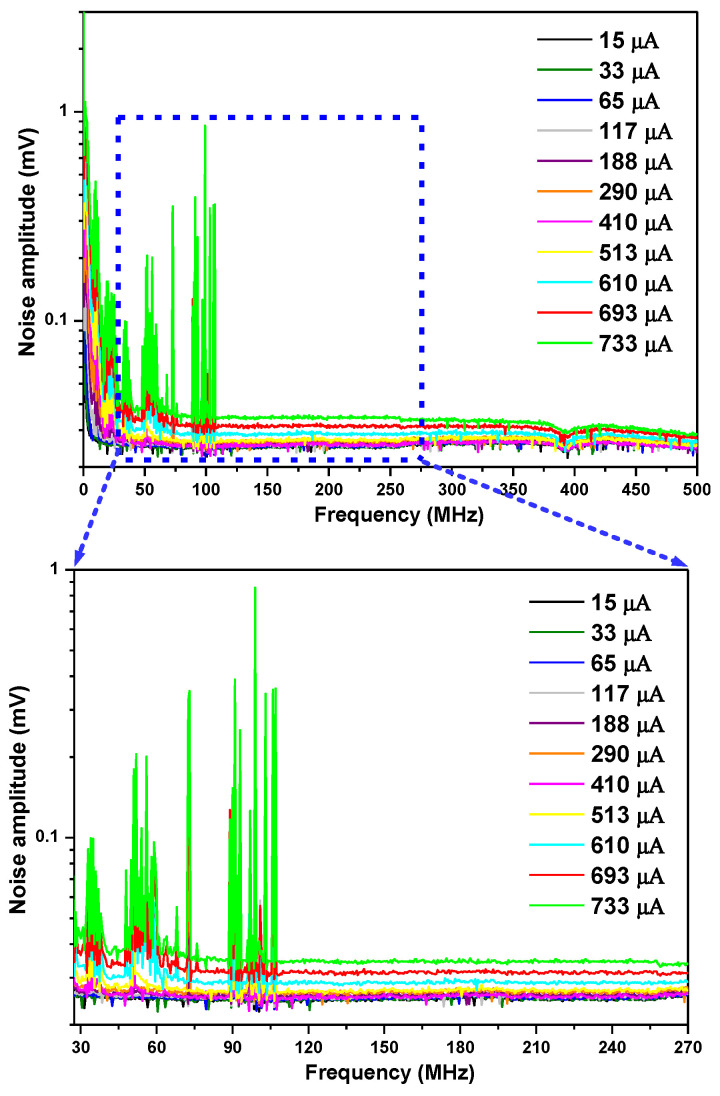
(**upper**) noise density spectra from the RF output of the photomixer at different LO power-induced photocurrents measured with a signal analyzer; (**lower**) zoom of the upper figure in the pass-band of 27.5–270 MHz.

**Figure 5 sensors-23-00978-f005:**
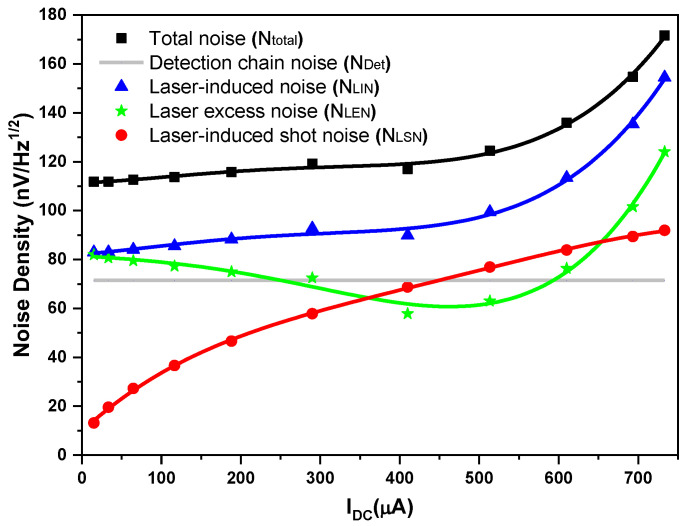
Plots of measured system total noise density *N_total_*, detection chain noise density *N_Det_*, laser-induced noise density *N_LIN_*, laser-induced shot-noise density *N_LSN_*, and laser excess noise density *N_LEN_* vs. LO photocurrents, respectively.

**Figure 6 sensors-23-00978-f006:**
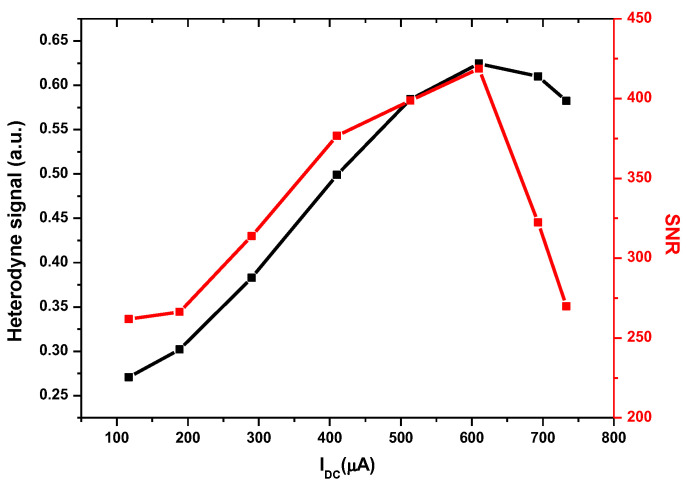
Evolution of heterodyne signal (black square) and SNR (red square) as a function of LO photocurrent.

**Figure 7 sensors-23-00978-f007:**
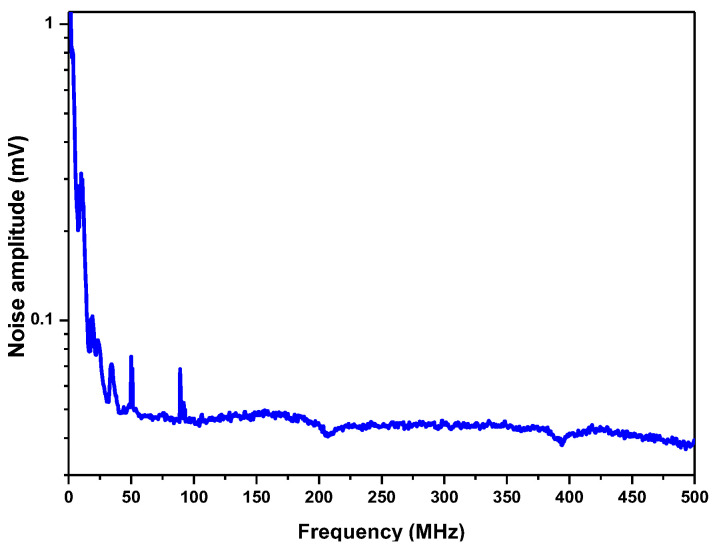
Noise amplitude spectrum from the RF output of the photomixer in the 1 kHz–500 MHz range.

**Figure 8 sensors-23-00978-f008:**
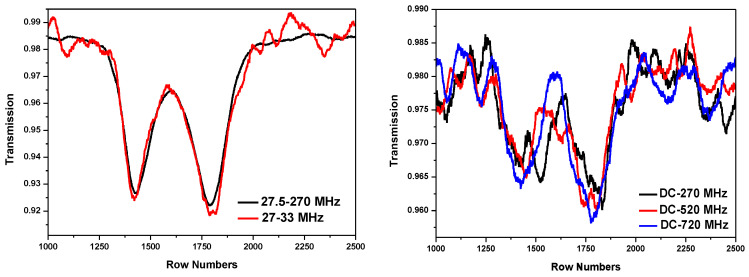
Heterodyne spectra of CH_4_ extracted with different RF filter pass-bands.

**Figure 9 sensors-23-00978-f009:**
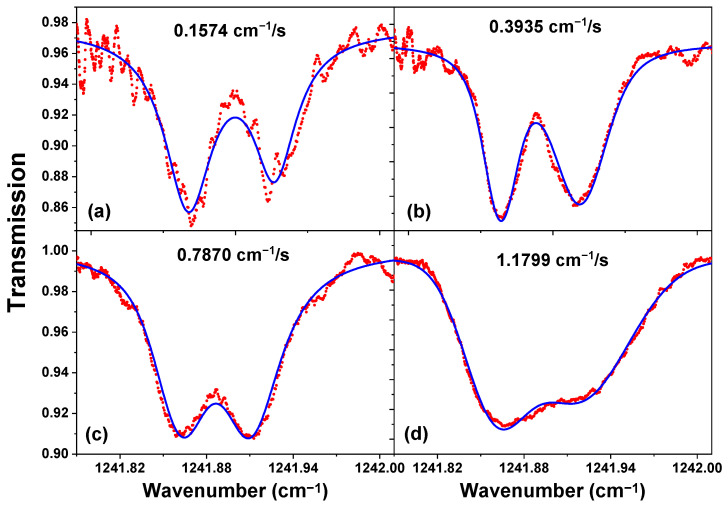
Heterodyne absorption spectra of CH_4_ (red) at scanning speeds of 0.1574 cm^−1^/s (**a**), 0.3935 cm^−1^/s (**b**), 0.7870 cm^−1^/s (**c**),and 1.1799 cm^−1^/s (**d**), respectively, fitted with Voigt profiles (blue).

**Table 1 sensors-23-00978-t001:** Noise density with different RF filter pass-bands.

Pass-Band (MHz)	DC-270	DC-520	DC-720	27–33	27.5–270
Noise density ($\mu V/\sqrt{Hz}$)	0.396	0.239	0.239	0.180	0.149

**Table 2 sensors-23-00978-t002:** Analysis of BB-based heterodyne spectra (Figure 9) of CH_4_ absorption with the used time constants (*τ*) of 1 ms at different scanning speeds of the LO frequency (*υ_sc_*). (Note: the halfwidth of the used CH_4_ line Δ*ν_HWHM_* at 280 mbar is 0.014 cm^−1^ [29]; 1 *σ*: Standard Deviation (SD)).

τ (ms)	υ_sc_ (cm^−1^/s)	Δ*T_scan_* (ms)	Δ*ν_b_* (cm^−1^)	Absorption Depth	δ*ν* (cm^−1^)	Fit Residual (1*σ*)
1	0.1574	89	0.038	0.1142	1.6 × 10^−4^	0.0091
	0.3935	36	0.042	0.1037	4.0 × 10^−4^	0.0043
	0.7870	18	0.048	0.0975	8.2 × 10^−4^	0.0032
	1.1799	12	0.079	0.0909	1.2 × 10^−3^	0.0021

## Data Availability

Not applicable.

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
