# Peer review of "Performance Characterization of a Fully Transportable Mid-Infrared Laser Heterodyne Radiometer (LHR)"

_sensors, 2023, doi:10.3390/s23020978_

Round 1
Reviewer 1 Report
The paper by F. Shen et al. reports the characterization and optimization of a fully transportable laser heterodyne radiometer (LHR) with a flexible polycrystalline mid-infrared fiber coupling system. The paper is well written, and the technical contents are sound and reasonable. I recommend its publication on Sensors provided that the authors address a few comments I enumerate below.
1. Current version of LHR operates around 8 μm. Please introduce the consideration for choosing this wavelength. Can it operate at other mid-IR wavelengths?
2. The propagating direction of all light beams should be marked in Fig. 1 for helping readers tracing the light.
3. Why the difference between the two curves in Fig. 2 changes with time? Please be more specific.
Author Response
Dear reviewer,
We thank you very much for your careful review works which allows us to significantly improve the quality of this paper. We revised the manuscript based on your suggestions and comments. please check the attachment.

Reviewer 2 Report
Reviewer’s comments on ‘’ Performance Characterization of a Fully Transportable Mid-infrared Laser Heterodyne Radiometer (LHR)’’ by Shen et al.
The paper describes the set-up and characterization of a radiometer to be used to measure trace gases in the atmosphere. The apparatus proposes to use absorption of sunlight to measure absorption spectra for molecular gases. A sample methane spectrum obtained using a 280 mbar gas cell and 1000 K black body source is presented as a test of the system. Noise and signal strength measurements of heterodyne detection are given.
The paper is usually detailed in the presentation of the constructed equipment. Nevertherless, there are some omissions that should be rectified before publication:
(1) In the first paragraph, the authors stress the importance of a reliable local oscillator for the heterodyne mixing. However, apart from giving the model name and number (p2, line 80), the properties of the local oscillator are not described (eg. Power, frequency, stability etc). The author should describe properties of the local oscillator in more detail.
(2) The presentation of the solar tracking and position data (p3 lines, lines 115 – 130 and figure 2) seems to be out-of-place and may confuse that a complete solar tracking system is being described, when, in fact, all results obtain to methane in a gas cell with a laboratory quasi-black-body source. I suggest that the authors remove this data from the paper.
(3) The testing of the system was undertaken with a 1000 K black-body source, but will ultimately be used with solar radiation (temperature 5800 K). The authors should comment in the introduction or elsewhere as to whether this temperature difference is significant when the apparatus is used with solar radiation.
(4) There is no indication that the system will be sufficiently sensitive to atmospheric levels of gases. The areal density of methane in the gas cell is probably high compared to eg levels of methane over 100s m in the atmosphere. The authors should give an indication of the sensitivity in the atmosphere of the detection system.
Author Response

(The authors gave the same response as above.)
